# VCAM-1 Target in Non-Invasive Imaging for the Detection of Atherosclerotic Plaques

**DOI:** 10.3390/biology9110368

**Published:** 2020-10-29

**Authors:** Kathleen Thayse, Nadège Kindt, Sophie Laurent, Stéphane Carlier

**Affiliations:** 1Laboratory of Cardiology, Faculty of Medicine and Pharmacy, Université de Mons, 7000 Mons, Belgium; kathleen.thayse@umons.ac.be; 2Laboratory of Oncology and Experimental Surgery, Institut Jules Bordet, Université Libre de Bruxelles, 1000 Brussels, Belgium; nadege.kindt@bordet.be; 3Laboratory of General, Organic and Biomedical Chemistry, Faculty of Medicine and Pharmacy, Université de Mons, 7000 Mons, Belgium; sophie.laurent@umons.ac.be

**Keywords:** VCAM-1, atherosclerosis, molecular imaging

## Abstract

**Simple Summary:**

Cardiovascular diseases are the first cause of morbimortality worldwide. They are mainly caused by atherosclerosis, with progressive plaque formation in the arterial wall. In this context, several imaging techniques have been developed to screen, detect and quantify atherosclerosis. Early screening improves primary prevention and promotes the prescription of adequate medication before adverse clinical events. In this review, we focus on the imaging of vascular cell adhesion molecule-1, an adhesion molecule involved in the first stages of the development of atherosclerosis. This molecule could therefore be a promising target to detect early atherosclerosis non-invasively. Potential clinical applications are critically discussed.

**Abstract:**

Atherosclerosis is a progressive chronic arterial disease characterised by atheromatous plaque formation in the intima of the arterial wall. Several invasive and non-invasive imaging techniques have been developed to detect and characterise atherosclerosis in the vessel wall: anatomic/structural imaging, functional imaging and molecular imaging. In molecular imaging, vascular cell adhesion molecule-1 (VCAM-1) is a promising target for the non-invasive detection of atherosclerosis and for the assessment of novel antiatherogenic treatments. VCAM-1 is an adhesion molecule expressed on the activated endothelial surface that binds leucocyte ligands and therefore promotes leucocyte adhesion and transendothelial migration. Hence, for several years, there has been an increase in molecular imaging methods for detecting VCAM-1 in MRI, PET, SPECT, optical imaging and ultrasound. The use of microparticles of iron oxide (MPIO), ultrasmall superparamagnetic iron oxide (USPIO), microbubbles, echogenic immunoliposomes, peptides, nanobodies and other nanoparticles has been described. However, these approaches have been tested in animal models, and the remaining challenge is bench-to-bedside development and clinical applicability.

## 1. Introduction

Cardiovascular diseases (CVDs) are the first cause of morbimortality worldwide [1,2]. CVDs cause almost a third of all the deaths worldwide and 43% of all non-communicable disease deaths, which represents just under double the number of deaths that are caused by cancer [3]. In the latest available European data, CVDs even cause up to 47% of all deaths [4]. The ageing and growth of populations have led to an increase in cardiovascular mortality over the years [5]. However, while age-standardised death rates have been decreasing since 1987 due to progress in treatment and primary prevention [6,7], there has been a recent increase in this rate [4].

Atherosclerosis is the main cause of CVDs, and its major clinical manifestations are coronary artery disease (CAD), stroke and peripheral arterial disease [8] Atherosclerosis is a progressive chronic arterial disease characterised by atheromatous plaque formation in the intima through inflammation, immune cell activation, lipid accumulation, necrosis and fibrosis [9]. Several risk factors increase the likelihood of developing atherosclerosis. Smoking, high blood pressure, dyslipidaemia, diabetes and obesity [8] are several of the modifiable risk factors.

It is important to control risk factors adequately to prevent the development of atherosclerosis, which causes cardiovascular events [10]. The Screening for Heart Attack Prevention and Education (SHAPE) task force has demonstrated that even low risk patients can suffer from cardiovascular problems [11]. It is essential to develop non-invasive complementary tests to detect these patients early, instead of the coronary angiogram, which is the current gold standard for assessing coronary artery stenosis, but which is an invasive test that cannot visualise the plaque burden. Early screening permits the enhancement of primary prevention and the initiation of adequate medication management before clinical events. In this review, we focus on the vascular cell adhesion molecule-1 (VCAM-1), a protein that can be a promising target to detect early and advanced atherosclerotic lesions by non-invasive imaging techniques.

### 1.1. Atherosclerosis

Endothelial dysfunction is the first step in the development of atherosclerosis. It is promoted by shear stress induced by turbulent blood flow, high blood pressure, smoking or intermittent hypoxia responsible for an imbalance of oxidative stress in apnoeic patients. The endothelial cells begin to secrete chemokines that recruit monocytes and T lymphocytes [12,13]. These chemokines have a specificity for leucocyte cell types that express chemokine receptors [14]. The attracted leucocytes use the three following steps before transendothelial migration: (1) rolling adhesion, (2) activation and (3) arrest by firm adhesion. During the rolling adhesion step, they develop transient weak interactions with selectins on activated endothelial cells. Then, the leucocytes that slow down produce cytokines, such as TNF alpha, which promote the production of more cytokines, but also the overexpression of leucocyte-adhesion molecules on the surface of endothelial cells [15]. The activated endothelium also secretes chemokines that increase the leucocyte integrins’ affinity for vascular adhesion molecules [13]. The leucocytes integrins α4β1 (very late antigen-4 (VLA-4)), αLβ2 (LFA-2, CD11a/CD18) and αMβ2 (Mac-1, CD11b/CD18) bind, with high affinity, to the adhesion molecule expressed on the endothelial cells, which initiates intracellular signalling in the endothelial cells, resulting in junctional disruption, and permits the transendothelial migration of the leucocytes [15].

Modifications of low-density lipoproteins (LDL), and in particular their oxidation, play a major role in the atherosclerotic process. LDL can be oxidised by oxygen reactive species (ROS) produced by endothelial cells or can be oxidised by lipoxygenases or myeloperoxidases produced by neutrophils and activated blood monocytes.

After endothelial dysfunction, the monocytes migrate in the subintimal space and become macrophages that phagocyte ox-LDL after binding the scavenger receptors. The accumulation of the ox-LDL in macrophages induces their transformation into foam cells, which have pro-inflammatory effects by secreting cytokines and growth factors (IL1*β*, IFN-*γ*, TNF*α*, M-CSF). Foam cells progress to apoptosis and form a necrotic core. The smooth muscle cells of the media also migrate into the intima and proliferate due to growth factors secreted by macrophages, endothelial cells and T lymphocytes. These smooth muscle cells can incorporate ox-LDL and secrete extracellular matrix proteins, causing the formation of a fibrous cap. Furthermore, the lack of fibrinolysis that contributes to atherogenesis is a risk factor for thrombotic complications. Plaque rupture also promotes the development of a pro-thrombotic environment [12,13].

Different types of lesions have been defined according to the stage of development, from Grade I (characterised by the presence of isolated foamy macrophages), to Grades V (atheromatous plaque) and VI (rupture/erosion leading to thromboembolic events) [16] (Figure 1).

### 1.2. Vascular Cell Adhesion Molecule-1 

The main adhesion molecules expressed by the activated endothelium are E-selectin, intracellular adhesion molecule-1 (ICAM-1) and vascular cell adhesion molecule-1 (VCAM-1) [17,18]. They are responsible for the adhesion and migration of the monocytes and the lymphocytes. However, their prevalence in atherosclerotic lesions is different: E-selectin is rarely expressed in atheromatous plaques (14% of cases); ICAM-1 is present in 46% of atherosclerotic lesions; and VCAM-1 is the most prevalent adhesion molecule in atherosclerosis (82%). Moreover, VCAM-1 has a major role in the initiation of atherosclerosis and is present in early lesions [19]. Therefore, VCAM-1 is a good target for the detection of existing atherosclerotic lesions, as well as for the early detection of an activated endothelium at-risk of developing plaques.

VCAM-1, also called CD106, is an 110kDa transmembrane glycoprotein of the immunoglobulin superfamily. It is composed of an extracellular immunoglobulin-like domain with several immunoglobulin (Ig)-like domains containing disulphide-linked loops, one single type I transmembrane domain and one short cytoplasmic part. The number of Ig-like domains is caused by the alternative splicing of mRNA from a single VCAM-1 gene to form two isoforms [20] and differs between species. Human VCAM-1 has two splice variants: a predominant form with seven immunoglobulin-like domains or a six-domain form that lacks domain 4, whereas in mice, for example, VCAM-1 can contain three or seven Ig-like domains [21]. 

The expression of VCAM-1 on a cell’s surface is induced by pro-inflammatory cytokines (like TNF-alpha) [22], 25-hydroxycholesterol [23], low density ox-LDL [24], high glucose exposure [25], toll like receptor agonists, oxidative stress [26,27] and shear stress [28]. It leads to the activation of many intracellular pathways that regulate the activation of VCAM-1: nuclear factor kappa B (NFκB) [29,30], SP-1 [26], activating protein-1 (AP-1) [31] and interferon regulatory factor-1 [32] pathways. 

VCAM-1 is expressed principally on the surface of activated endothelial cells [33], but is also found on other cells in inflammatory conditions or chronic pathologies: macrophages, follicular dendritic cells [34], fibroblasts [35], mesenchymal stem cells [36], myoblasts, oocytes, Kupffer cells, Sertoli cells and cancer cells [37]. On the endothelial surface, VCAM-1 is colocalised with proteins that promote its function in the first stage of atherosclerosis. It is included in a tetraspanin-enriched microdomain (TEM) that contains ICAM-1, CD9, CD81 and CD151, which bind to each other via their extracellular domains [38]. For example, tetraspanin CD81 upregulates integrin affinity to VCAM-1 for leucocyte rolling and arrest [39], and tetraspanin 151 promotes the surface expression of VCAM-1 and its adhesive function for leucocyte transendothelial migration [2,36]. Other proteins, ezrin and moesin, connect VCAM-1 to the membrane-actin cytoskeleton, allowing immune cell adhesion [40].

Once expressed on the endothelial cell surface, VCAM-1 can bind to several ligands on leucocytes: principally α4β1 integrin [41], but also α4β7 integrin [42], αdβ2 integrin [43], galectin-3 [44] and osteonectin [45]. The α4-integrins bind to two domains of seven Ig-like domain forms of VCAM-1, the first (D1) and the fourth (D4) domain, and this binding is regulated by the activation state of the integrins. The major ligand of VCAM-1, α4β1 integrin (also called VLA-4, or CD49d/CD29) expressed on leucocytes, plays an important role in the rolling adhesion and the firm adhesion of leucocytes before their transmigration [46,47]. Non–activated α4β1 integrin binds to the acidic motif QIDSPL within D1 of VCAM-1 under regulation by CD24 expression [48], but α4β1 integrin requires higher affinity activation by divalent cations or chemokines [48] for binding to domain 4 [49] (Figure 2). 

The binding of ligands to VCAM-1 activates calcium flow and Ras-related C3 botulinum toxin substrate 1 (Rac1), which induces two different pathways. One pathway begins with the activation of nicotinamide adenine dinucleotide phosphate oxidase 2 (NOX2). NOX2 produces reactive oxygen species (ROSs), which activate matrix metalloproteinases (MMPs) and protein kinase Cα (PKCα). Activated PKCα leads to the activation of protein tyrosine phosphatase 1B (PTP1B), which has a role in the disruption of endothelial cells junctions in the transendothelial migration of leucocytes. Rac1 induces another pathway, the Rac1-p21-activated protein kinase (PAK)-myosin light chain (MLC) pathway, which stimulates the formation of actin fibres. All these pathways lead to cytoskeletal remodelling, the weakening of the intercellular junctions between endothelial cells and, thereby, facilitate leucocyte migration [50] (Figure 3).

Moreover, VCAM-1 can be released from the endothelial surface through proteolytic cleavage by disintegrin metalloproteinases, principally ADAM17 (also called tumour necrosis factor-alpha-converting enzyme (TACE)) [51,52] and to a lesser extent ADAM8 [53] and ADAM9 [54]. This juxtamembrane cleavage releases a 100 kDA soluble protein, sVCAM-1, into the plasma [14]. This soluble form of VCAM-1 stimulates leucocyte chemotaxis by binding high affinity α4β1 integrin [55]. Thereby, sVCAM1 is a promising biomarker for atherosclerotic diseases such as coronary artery disease [56], but higher levels of sVCAM-1 are also found in other diseases: type 1 diabetes mellitus [57], inflammatory bowel disease [58] and acute bronchiolitis in children [59]. It has also a predictive interest for diseases such as cardiac allograft vasculopathy in heart transplants [60], atrial fibrillation after coronary artery bypass surgery [61], recurrent ischemic stroke [62] and post-operative recurrence in colorectal cancer [63]. 

## 2. Imaging Techniques

### 2.1. Imaging of Atherosclerosis

Several invasive and non-invasive imaging techniques have been developed to detect and characterise atherosclerosis in the vessel wall: anatomic/structural imaging, functional imaging and molecular imaging.

Invasive angiography is the gold standard imaging technique to detect and assess the severity of advanced atherosclerotic lesions leading to coronary stenosis and to guide revascularisation. It involves the intubation of the coronary ostia with a catheter introduced via a peripheral arterial sheath, in order to inject a radio-opaque contrast under X-ray fluoroscopy [64]. It is invasive and exposes the patient to: (1) ionising radiation, which is of concern as it has been established that significant radiation exposure is associated with an increased risk of cancer, and (2) iodinated contrast agents, to which the patient may be allergic and that can cause nephrotoxicity. Current research aims to develop non-invasive imaging techniques for atherosclerosis diagnosis that have as few adverse events as possible for the patient.

Non-invasive angiography is possible with two anatomic atherosclerosis techniques: coronary computed tomographic angiography (CCTA) and magnetic resonance angiography (MRA) combined with ECG gating to account for cardiac movement. CCTA is better than MRA for coronary imaging and is indicated for symptomatic patients with low-to-moderate pre-test probability of coronary artery disease (CAD). As reported in the Scottish Computed Tomography of the Heart (SCOT-HEART) trial, computed tomographic angiography (CTA) is useful for the evaluation of patients with stable chest pain and leads to the use of consequent appropriate therapies, which results in fewer five year clinical outcomes than patients treated with standard care alone [65]. However, when there is a high pre-test probability of CAD, invasive angiography is preferred [64].

Other structural imaging techniques enable the characterisation of plaque on the vessel wall. Plaque morphology and composition are predictors for the risk of plaque rupture and, therefore, adverse cardiac events. Coronary plaque composition can be assessed during invasive angiography by intravascular coronary imaging using ultrasound (IVUS), optical coherence tomography (OCT) and near infrared-spectroscopy (NIRS). Non-invasive CCTA and magnetic resonance imaging (MRI) can also provide some plaque characterisation [64].

However, luminal stenosis of the coronary artery does not always reliably correlate with hemodynamic obstruction. Non-invasive (stress imaging) or invasive functional imaging tests (pressure-wire assessment during angiography) can determine hemodynamically significant lesions. Non-invasive functional imaging tests include stress echocardiography, cardiac MR with stress perfusion and nuclear myocardial perfusion scanning with single photon emission computed tomography (SPECT) or positron emission tomography (PET) [64].

In addition to structural and functional imaging techniques, molecular imaging is the specific imaging of cellular and molecular biomarkers. It helps us to understand the pathobiology of atherosclerosis and its clinical consequences using molecular imaging probes in PET, SPECT, CT, MRI, US and optical imaging. These probes enable us to quantify vascular inflammation, early calcification, plaque hypoxia and neoangiogenesis. To enhance the specific target of atherosclerosis, the probes can be linked with a specific ligand, such as peptides, nanobodies or antibodies directed against the molecular target. Probes can also be labelled with other imaging tracers to create multimodal imaging probes. Multimodal imaging enables us to combine the information and advantages from different imaging techniques [64]. For example, Nahrendorf’s team created a tri-modal nanoparticle to detect macrophages in atherosclerotic plaques using PET, MR and optical imaging [66].

In nuclear imaging, PET and SPECT, the administered probes are radiotracers used for inflammation, microcalcification, hypoxia and neoangiogenesis detection. The most commonly used radiotracer in PET imaging is [^18^F]fluorodeoxyglucose (FDG), a radio-labelled glucose analogue. It is taken up by cells that metabolise glucose, and its intracellular accumulation is used as a biomarker of metabolic activity. In atherosclerosis, it reflects the increased activity of macrophages and, to a lesser extent, other immune cell types. However, [^18^F]FDG is a non-specific marker of inflammation in atherosclerosis, also influenced by hypoxia, microcirculation and uptake by the myocardial muscle. It has been used in patients, but no difference in culprit and non-culprit coronary lesions could be demonstrated, due to shadowing from myocardial uptake [67]. Other PET tracers have been evaluated to avoid these limitations [64]. [^18^F]sodium fluoride (NaF) can detect high-risk plaques as demonstrated in the same study that failed to demonstrate a clinical value for [^18^F]FDG imaging [68]. It can also detect microcalcifications [68] in the thin cap of fibroatheroma lesions, which are known to favour plaque rupture [69], and independently predicts myocardial infarction [70], which makes [^18^F]NaF interesting for stratifying the risk in atherosclerosis. There are also PET tracers for the detection of hypoxia, such as [^18^F]fluoromisonidazole [70], and for the detection of neoangiogenesis, such as [^18^F]galacto-RGD, which targets integrin ανβ3 expression [71]. All these radiotracers enable us to study atherosclerosis in PET, which is an interesting imaging technique due to its very high sensitivity. However, this technique suffers from limitations, such as limited spatial resolution and the exposure to radiotracers and ionising radiation [72,73]. Some contrast agents have also been tested in CT, another irradiating imaging technique. These agents, iodinated nanoparticulate contrast agent N1177 and gold-core high-density lipoprotein particles, are able to detect activated macrophage infiltration in atherosclerotic plaques [74,75], which improves the identification of high-risk coronary atherosclerotic plaques compared to CCTA. Indeed, they bypass the main limitation of CCTA: the high variability of luminal enhancement, which makes it difficult to measure atherosclerotic plaque densities in order to characterise them. Nevertheless, this does not solve the general disadvantages of CT, which are radiation exposure, spatial resolution and soft tissue contrast [73].

Among non-irradiating imaging techniques, MRI is the most extensively studied technique due to its high spatial resolution and its excellent soft tissue contrast [76]. However, its clinical use is limited by its incompatibility with metallic medical implants [73]. Contrast agents used in molecular MRI are T1-shortening agents based on gadolinium (Gd) chelates and T2-shortening agents based on iron oxide particles [77]. Because of the renal toxicity of Gd-based contrast agents, iron oxides have received more attention [73], including micrometre-sized particles of iron oxide (MPIOs; 0.9–8 µm in diameter), superparamagnetic iron oxide nanoparticles (SPIONs; 60–250 nm in diameter) and ultra-small superparamagnetic particles of iron oxides (USPIOs; 20–50 nm in diameter) [78]. Their spontaneous phagocytosis by macrophages enables us to detect macrophage activity in atherosclerotic plaques, as demonstrated in preclinical studies [72,79,80,81,82,83,84], but the MRI contrast agents can also be conjugated to a specific ligand to visualise specific sequences of the atherosclerotic process: endothelial dysfunction, endothelial cell activation, inflammation, angiogenesis, apoptosis, platelet activation and thrombosis [85]. The advantages of MPIOs are to deliver an excellent contrast effect through providing a high payload of iron oxide [79,86] and to have a rapid clearance from circulation when they are not bound to the target, which minimises background blood phase contrast and improves plaque visualisation [87]. Unlike the MPIOs, USPIOs circulate longer because they are not well recognised by phagocytic cells, mostly those of the liver and spleen, due to their small size [88], and are therefore more prone to extravasation. Their longer intravascular half-life requires that the postcontrast imaging time be rather long to allow linkage or uptake of USPIOs, and their potential extravasation induces a higher background signal [89]. Another disadvantage highlighted in animal studies is the need for high administration dose to detect atherosclerotic plaques [89]. However, it has been shown that the dose administered in human studies is much lower than in experimental animal models [89,90,91]. Schmitz et al. studied the detection of lymph node metastases by USPIO in patients with gynaecological and urological malignancies and, following the analysis of the MR images, they retrospectively found the accumulation of USPIO in human atherosclerotic plaques in the wall of the abdominal aorta or of the iliac vessels [90]. Following this discovery, Kooi et al. decided to perform a successful in vivo detection of USPIOs in the atherosclerotic plaques of the carotid arteries of patients before they underwent carotid endarterectomy, which subsequently allowed ex vivo histological analysis of the samples [89]. Unfortunately, there was no correlation between the degree of USPIO-enhanced MR-defined inflammation and the severity of luminal stenosis in asymptomatic carotid plaques [91]. However, the Atorvastatin Therapy Effects on Reduction of Macrophage Activity (ATHEROMA) study demonstrated that a high dose of atorvastatin over a three month period in patients with carotid stenosis induced a significant reduction in USPIO-enhanced MRI-defined plaque inflammation [92].

Contrast agents used in molecular MRI are attractive, but do not always provide sufficient contrast enhancement to visualise atherosclerotic plaques. Therefore, nanoparticles (from 1 to 100 nm in diameter) have been developed to carry a substantial payload of MRI contrast agent and a specific target against atherosclerosis [86]. These nanoparticles are interesting because of the great potential for tuning their structure: various sizes and shapes of nanoparticles can be synthesised to improve their vascular targeting efficiency in the blood circulation. Spherical shapes are less optimal for vascular imaging than asymmetric oblong shapes because of their rapid macrophage internalisation and the low number of ligands available for binding. Non-spherical nanoparticles are then preferred due to their superior endothelial targeting, reduced macrophage uptake and better avidity for the ligands [93]. Moreover, the nanoparticles may be labelled with fluorophores for preclinical tests with non-invasive and intravascular near-infrared fluorescence (NIRF) imaging, confocal microscopy and flow cytometry [64]. Nanoparticles are used in diagnostic, as well as therapeutic applications for atherosclerosis [72].

Other non-irradiating imaging techniques, such as contrast-enhanced ultrasound (CEU), which uses targeted microbubbles and non-invasive NIRF imaging, have several interesting advantages, such as cost and the speed of acquisition [77,94]. On the one hand, CEU uses microbubbles that are low in solubility and diffusion inert gases surrounded by a shell of albumin, lipid surfactants or biocompatible polymers. They are small, less than 5 µm in diameter, in order to cross the pulmonary and systemic capillary circulation to reach the systemic circulation and resonate in an ultrasound field at the frequencies used in diagnostic sonography. Their limitations are their short lifespan and their inability to access targets outside the vascular space [94]. On the other hand, non-invasive NIRF imaging uses fluorescent probes emitting in the near-infrared window. These probes can detect atheroma if they are linked with a specific ligand. However, the penetration depth of the light is limited to a few mm, a maximum of 1 cm of tissue depth, which unfortunately restricts its use to mice and means it is not applicable in humans [77].

### 2.2. Imaging of VCAM-1

Atherosclerosis is associated with increased endothelial cell (EC) expression of adhesion molecules, especially VCAM-1. The endothelial location of VCAM-1 makes it easily accessible to intravascular imaging agents [78]. Hence, for several years, there has been an increase in molecular imaging methods for detecting VCAM-1 in MRI, PET, SPECT, optical imaging and ultrasound. These methods consist of the use of microparticles of iron oxide (MPIO), ultrasmall superparamagnetic iron oxide (USPIO), microbubbles, echogenic immunoliposomes, radiotracers, nanoparticles and biofunctional probes. These contrast agents are linked to peptides, antibodies or nanobodies that specifically target VCAM-1 (Table 1). 

For specific targeting of VCAM-1 in MR imaging, MPIO and USPIO are linked with anti-VCAM-1 ligands. McAteer et al. demonstrated that anti-VCAM-1 antibody conjugated to MPIO allowed the localisation of atherosclerotic lesions in the aortic root of ApoE knockout mice. They also generated a dual-MPIO for binding VCAM-1 and P-selectin in the same murine model, which demonstrated a better targeting for aortic atherosclerotic plaques than single ligand-targeted MPIO [95]. In addition to their high specificity, the dual-MPIO has other advantages, such as its sensitivity due to the high iron content of MPIOs, its swift clearance from blood circulation and their rapid binding. This makes dual-MPIO an interesting molecular imaging tool for clinical use, as suggested by Chan et al. [87]. Chan et al. showed that dual-MPIO enabled the detection of vulnerable inflamed plaques in mouse carotid arteries by MRI (Figure 4) and therefore can potentially allow risk stratification of individual patients and identification of patients at high risk of cerebrovascular diseases in order to manage early prevention and adequate treatment. The main limitation is the high dose of dual-MPIO administered in the study compared to the usual dose of nontargeted MRI contrast agent used clinically in human oncology. Otherwise, the specificity of dual-MPIO could be further increased in the future by replacing the two separate antibodies that target VCAM-1 and P-selectin with bispecific antibodies that simultaneously target both VCAM-1 and P-selectin [87].

Otherwise, recently, Rucher et al. coupled MPIO-VCAM-1 magnetic resonance with [^18^F]NaF PET in an ApoE^−/−^ mouse model with chronic renal failure in order to identify both endothelial activation and microcalcifications [96]. This preclinical investigation opens new perspectives for the evaluation of the dynamic process of atherosclerosis by helping understand the relationship between inflammation, early mineralisation, which is known to detect high risk plaques, and vascular remodelling observed in this murine model. 

Concerning USPIO, the teams of Burtea and Michalska developed a VCAM-1 peptide conjugated USPIO (USPIO-R832 and P03011, respectively) to identify atherosclerotic plaques in the ApoE^−/−^ mouse model by MRI and to contribute to the detection of vulnerable plaques [97,98]. Thanks to targeted USPIO, these teams tried to limit the disadvantages of USPIO by reducing the administered dose and blood half-life, but this latter should be even shorter, in order to limit the background signal as much as possible. Indeed, targeted USPIO has higher sensitivity and specificity to detect inflamed regions than untargeted USPIO due to differences in their size and surface properties. As explained previously, untargeted USPIO is taken up by plaque macrophages, but its accumulation must be sufficient to be visualised by MRI, which explains the higher administration doses and blood half-life than targeted USPIO [98].

In order to have better contrast enhancement than simple MRI agents, nanoparticles labelled with an MR contrast agent and anti-VCAM-1 ligand were developed. In 2013, Bruckman et al. generated tubular nanoparticles derived from the tobacco mosaic virus (TMV) and labelled with an optical probe, Gd(DOTA) MR contrast agent and oligopeptides specific to VCAM-1. These VCAM-TMV based probes were injected into ApoE^−/−^ mice fed on a high fat/cholesterol diet and demonstrated an accumulation of VCAM-TMV in the atherosclerotic lesions of the aorta in vivo by MRI (Figure 5) and ex vivo by fluorescence analysis [99]. The principal advantages for the potential clinical use of TMV based contrast agents are their short circulation times and rapid tissue clearance, which should avoid gadolinium renal toxicity. Likewise, Aanei and co-workers developed an icosahedral nanocarrier platform from genome-free MS2 viral capsids (27 nm) attached to VCAM antibodies. They observed that VCAM-MS2 particles were localised in atherosclerotic plaques, including the small plaques that formed at the branching of arteries from the main aorta [100]. Otherwise, Kelly et al. developed an iron oxide fluorescent nanoparticle conjugated to a phage display-derived peptide (VHSPNKK) that detects VCAM-1 expression in atherosclerotic lesions in ApoE^−/−^ mice in vivo by MRI and intravital confocal microscopy and ex vivo by fluorescence imaging [101]. This multimodal nanoparticle (with VHPKQHR, homologous to the previous peptide) also quantified the inflammatory response after statin treatment by detecting the pharmacotherapy-induced reduction in VCAM-1 expression [102]. It is interesting because of its high specificity, its improved in vivo pharmacokinetics and its long detection time by imaging techniques, which assumes the ability of the peptide to shuttle large molecules across the plasma membrane, therefore making it a potential diagnostic or developmental tool for novel targeted therapeutics.

Several studies have highlighted the use of CEU molecular imaging of VCAM-1 to evaluate the degree of inflammation in atherosclerosis. Indeed, Kaufmann et al. demonstrated that VCAM-1-targeted microbubble attachment was specific to the severity of inflammation in atherosclerosis depending on the in vivo model used: the signal enhancement was much greater in ApoE^−/−^ mice on a hypercholesterolemic diet than in ApoE^−/−^ on a chow diet and was low or absent in wild-type mice on a chow diet [103]. However, microbubbles tend to remain close to the axial centre of blood vessels, which is a disadvantage for targeting atherosclerosis in larger vessels, and their bonding with VCAM-1 is influenced by shear stress forces. To overcome these limitations, VCAM-1-targeted microbubbles have been coupled to magnetic streptavidin, which allows them to be manipulated with a static magnetic field. These magnetic microbubbles were more sensitive in detecting early atherosclerotic lesions in ApoE^−/−^ mice on a hypercholesterolemic diet than nonmagnetic microbubbles. Unfortunately, the use of magnetic microbubbles in clinical use needs further studies about the effects of a rotating magnetic field and a higher magnetic gradient field strength [104]. More recently, Yan et al. developed a triple targeted microbubble by integrating VCAM-1 and ICAM-1 antibodies and synthetic polymeric sialyl Lewis X onto microbubbles (MB_VIS_) to mimic leucocyte behaviour during inflammation. The ultrasound signal with this MB_VIS_ was two-fold higher than the dual-/single-target groups in early atherosclerosis mice. Furthermore, ApoE^−/−^ mice treated with atorvastatin displayed a significantly decreased ultrasound imaging signal compared to placebo-treated ApoE^−/−^ mice using MB_VIS_ [105], indicating that MB_VIS_ can be used as a tool to assess pharmacologic interventions. However, MB_VIS_ is not yet ready for clinical use because the avidin-biotin coupling used for the preparation of MB_VIS_ is unlikely to be applied in humans and because there are variations in blood shear stresses in humans compared to mice.

In addition to simple microbubbles, echogenic immunoliposomes (ELIPs) are other small (<1 µm) ultrasound contrast agents composed of phospholipid bilayers [107]. VCAM-1-targeted ELIPs detect and enhance intravascular ultrasound image of atheroma in Yucatan miniature pigs on a hypercholesterolemic diet after direct arterial injection [118]. Compared to ELIPs, microbubbles provide higher contrast enhancement and are injected intravenously to detect atherosclerosis [94].

For nuclear imaging purposes, Nahrendorf et al. developed a tetrameric VCAM-1-specific peptide labelled with fluorine-18, [^18^F]4V, that allowed non-invasive PET-CT imaging of atherosclerotic lesions expressing VCAM-1 in ApoE^−/−^ mice [108]. The team of Broisat described a peptide of the major histocompatibility complex-1 (MHC-1) molecule B2702 radiolabelled with isotopes iodine-123 and technetium-99 m that localised VCAM-1 in atherosclerotic plaques ex vivo in Watanabe heritable hyperlipidaemic rabbits [109] and in vivo in a murine model [110]. Despite the high sensitivity of nuclear imaging, none of these two probes are available for clinical use.

In addition to the usual anti-VCAM-1 ligands (peptide and monoclonal antibody), there is a specific anti-VCAM-1 nanobody, cAbVCAM-1-5, which is composed of small fragments (2.5 nm in diameter) derived from camelid heavy chain-only antibodies. Nanobodies have already proven their utility in the field of oncology as radiotracers [119] and, therefore, constitute an interesting subject of study in atherosclerosis. Therefore, Broisat et al. generated the technetium-99m-labeled anti-VCAM-1 nanobody ([^99m^Tc]Tc-cAbVCAM-1-5) recognising mouse and human homologues that successfully detect atherosclerosis by SPECT/CT imaging [111]. [^99m^Tc]Tc-cAbVCAM-1-5 was also used to demonstrate the antiatherogenic effect of statins [112] and ezetimibe [113] in a mouse model of atherosclerosis. It is a specific, sensitive and reproducible tool for the non-invasive imaging of atherosclerosis and the evaluation of novel antiatherogenic agents, which requires clinical validation. Moreover, for PET, Bala et al. labelled the anti-VCAM-1 nanobody with fluorine-18 ([^18^F]FB-cAbVCAM-1-5), which was shown to be specific for atherosclerotic lesions in ApoE^−/−^ mice when compared to control group C57Bl/6 mice [114]. [^18^F]FB-cAbVCAM-1-5 has a low myocardial uptake with a high lesion-to-myocardium ratio, which makes it promising for the detection of coronary atherosclerosis in patients compared to [^18^F]FDG, but further studies are needed for clinical use. Senders et al. developed a ^64^Cu-labelled cAbVCAM1-5 in a rabbit model of atherosclerosis whose uptake was negatively associated with the vessel wall area [116]. This result was consistent with the study of Iiyama et al. showing that the expression of VCAM-1 is expressed predominantly by the endothelium in early lesions [120]. Therefore, more investigations are needed for VCAM-1 nanobody radiotracers to detect such lesions. In order to increase spatial resolution and sensitivity using the latest β-CUBE PET technology, Bridoux et al. conjugated cAbVCAM1-5 with a novel restrained complexing agent (RESCA) allowing faster ^18^F-labelling via the Al^18^F-method, a process that has already had good results in a previous study concerning the targeting of T-cells by interleukin-2-derived radiotracers [121]. [^18^F]AlF(RESCA)-cAbVCAM1-5 accumulated in atherosclerotic lesions in the aortic arch of ApoE^−/−^ mice, but also unfortunately in the bone structure, which is probably due to the uptake of degradation products of the tracer by unclear biological processes (Figure 6). This bone uptake is difficult to avoid because the imaging must be performed at a precise time to ensure the lowest blood background signal and the highest probe signal, but this time correlates with the time of the formation of radio-metabolites [115].

Recently, cAbVCAM-1-5 has also been studied in CEU: Punjabi et al. coupled anti-VCAM-1 nanobodies with microbubbles (MBcAbVCAM-1-5) to detect VCAM-1 expression in an atherosclerotic mouse model and ex vivo in human endarterectomy specimens using non-invasive ultrasound imaging. This study revealed that MBcAbVCAM-1-5 detected VCAM-1 expression in early and established atherosclerotic lesions in vivo and ex vivo, but it is not yet ready for clinical translation because the atherosclerosis developed in the mouse model studied is not similar to human atherosclerosis [106].

A novel imaging modality with surface-enhanced Raman spectroscopy (SERS), using gold nanoparticles, is under development for the detection of atherosclerotic lesions. Indeed, antibody-functionalised gold nanoprobes have been designed to contain a unique Raman signature in order to detect ICAM-1, VCAM-1 and P-selectin by SERS and create multiplex imaging. This technique was used in vitro on endothelial cells, ex vivo on human coronary arteries and in vivo on a humanised mouse model and can indicate the state of vascular inflammation [117]. However, there are still many parameters to study for clinical use: synthesis under sterile conditions, use of substances already well known in humans, evaluation of immunogenicity and pharmacokinetics.

In summary, VCAM-1 imaging enables us to detect atherosclerotic plaques, stratify the risk and evaluate a new therapy in atherosclerosis in animal studies. Only a few studies [106,117] have demonstrated the efficiency of VCAM-1 targeting agents on ex vivo samples of human arteries.

## 3. Conclusions

Atherosclerosis is an important field of application of imaging techniques, especially molecular techniques. There is an increasing number of molecular imaging probes developed for non-invasive monitoring of atherosclerosis. VCAM-1 is a promising target to detect early and advanced atherosclerotic lesions, which are easily accessible to blood contrast agents due to their localisation on the endothelial surface. However, these approaches have, so far, been mainly tested in animal models, and the current challenge remains the assessment of their clinical applicability, safety and additional independent diagnostic and predictive value for events such as myocardial infarction.

## Figures and Tables

**Figure 1 biology-09-00368-f001:**
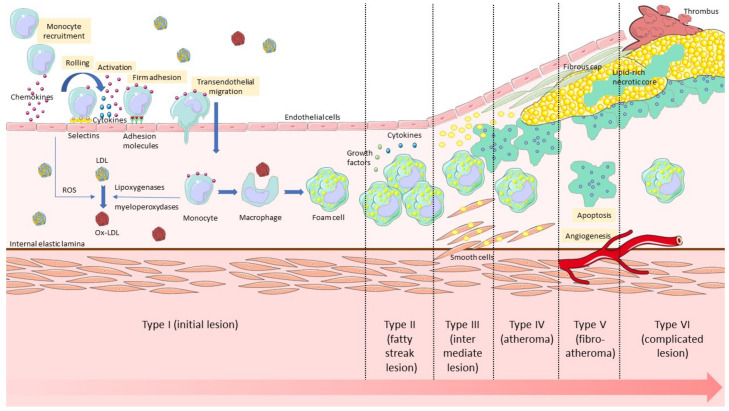
The different stages of the development of atherosclerosis, from endothelial dysfunction to plaque rupture with thrombosis.

**Figure 2 biology-09-00368-f002:**
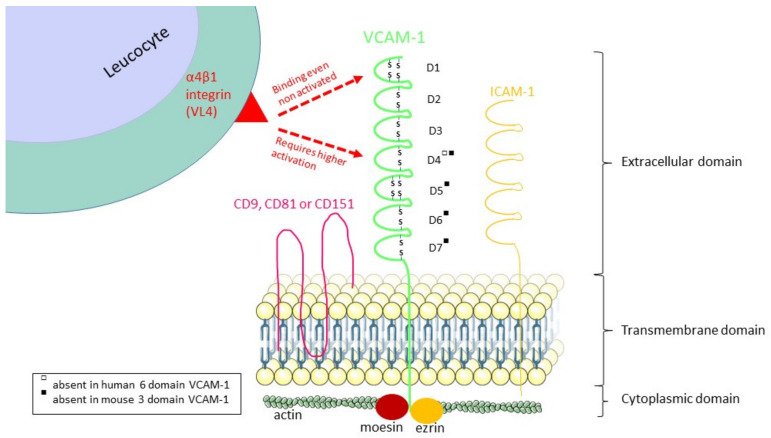
The structure of VCAM-1 and its α4β1 integrin binding regions. Human VCAM-1 has two splice variants with seven or six domains. Murine VCAM-1 has also two splice variants with seven or three domains. The α4β1 integrin binds to D1 of all the splice variants, but it binds to D4 if this domain is present and depending on the activation state of the α4β1 integrin.

**Figure 3 biology-09-00368-f003:**
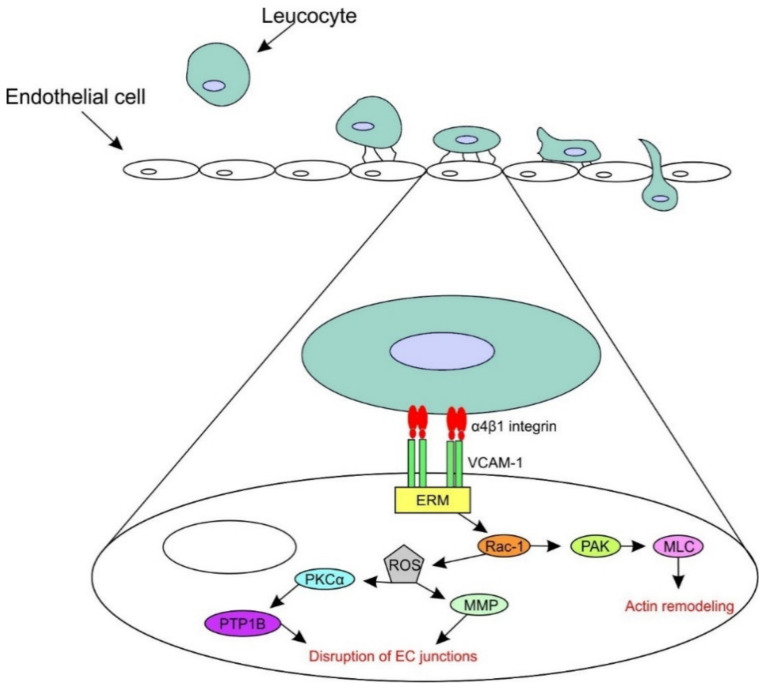
The signalling pathways initiated downstream of VCAM-1 binding with α4β1 integrin. The binding of the leucocyte to VCAM-1 and its complex ezrin-moesin (ERM) activates Rac-1. Rac-1 induces two pathways: (1) reactive oxygen species (ROSs), which activate matrix metalloproteinases (MMPs), protein kinase Cα (PKCα) and protein tyrosine phosphatase 1B (PTP1B) and which finally lead to the disruption of the endothelial cells’ (ECs) junctions and the (2) p21-activated protein kinase (PAK)-myosin light chain (MLC) pathway, which leads to actin remodelling.

**Figure 4 biology-09-00368-f004:**
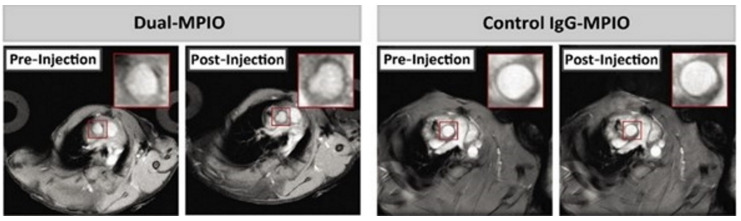
Images from the study of Chan et al. [87]. Detection of vulnerable inflamed plaques in mouse carotid arteries by in vivo MRI after administration of dual-MPIO (on the left) or control IgG-MPIO (on the right). In post-injection imaging after 2 h, the hypointensity signal in the aortic root was detected for dual-MPIO, but no change was observed for the control MPIO. Field of view: 25.6 mm × 25.6 mm. Acquisition matrix: 256 × 256. Reproduced with permission from Chan et al., Imaging Vulnerable Plaques by Targeting Inflammation in Atherosclerosis Using Fluorescent-Labeled Dual-Ligand Microparticles of Iron Oxide and Magnetic Resonance Imaging; published by J. Vasc. Surg., 2018.

**Figure 5 biology-09-00368-f005:**
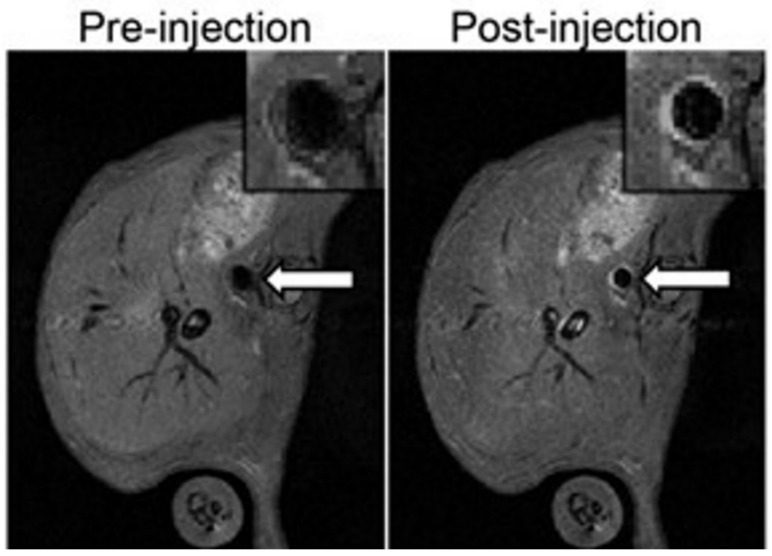
Images from the study of Bruckman et al. [99] Magnetic resonance imaging of the abdominal aorta of ApoE^−/−^ mice injected with VCAM-TMV demonstrated a positive enhancement of the arterial wall after 90 min of injection (on the right) compared to the images pre-injection (on the left). Field of view = 29.8 mm × 29.8 mm. Acquisition matrix: 256 × 256.

**Figure 6 biology-09-00368-f006:**
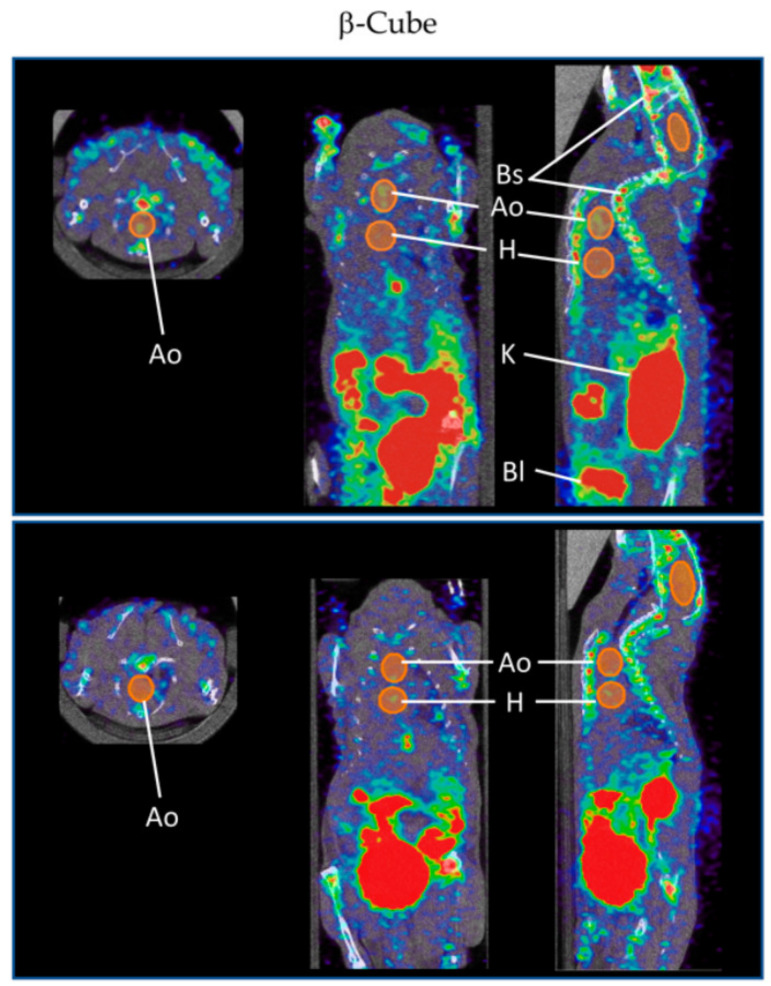
Images from the study of Bridoux et al. [115]. Detection of atherosclerotic lesions in the mouse aortic arch (Ao), near the heart (H), by in vivo β-CUBE imaging system after administration of [^18^F]AlF(RESCA)-cAbVCAM1-5 (top), but not after administration of control unlabelled cAbVCAM1-5 Nb (bottom). The kidneys (K), bladder (Bl) and bone structures (Bs) are also visible on the images. Voxel size of 0.4 and 0.5 mm. Open access article (Bridoux et al., Improved Detection of Molecular Markers of Atherosclerotic Plaques Using Sub-Millimeter PET Imaging; published by Molecules, 2020) distributed under the Creative Commons Attribution License.

**Table 1 biology-09-00368-t001:** Summary of the VCAM-1-targeted imaging agents. MPIO, microparticles of iron oxide; USPIO, ultrasmall superparamagnetic iron oxide; TMV, tobacco mosaic virus; ELIP, echogenic immunoliposomes.

Imaging Techniques	Contrast Agent	Targeting Moiety Used	Types of Study	References
MRI	MPIO	Monoclonal antibody (against VCAM-1 and P-selectin)	In vivo and ex vivo: ApoE^−/−^ on WD; In vivo and ex vivo: ApoE^−/−^ on standard or HD	[87,95]
MRI	MPIO	Antibody	In vivo: ApoE^−/−^ mouse model with chronic renal failure	[96]
MRI	USPIO	Peptide (R832)	In vivo: ApoE^−/−^ on WD	[97]
MRI	USPIO	Peptide (P03011)	In vivo and ex vivo: ApoE^−/−^ on WD	[98]
MRI	TMV-based probes with sulfo-Cy5-azide and Gd ions chelated with azido-mono amide- 1,4,7,10-tetraazacyclododecane-N-N′-N″-N‴-tetra acetic acid (Gd(DOTA))	Peptide	In vivo and ex vivo: ApoE^−/−^ on HD	[99]
MRI	MS2 particles with AlexaFluor680 (fluorescent dye with near infrared properties)	Antibody	Ex vivo: ApoE^−/−^ on HD	[100]
MRI + intravital confocal microscopy	Cross-linked iron oxide nanoparticle (CLIO)-Cy5.5	Peptide	In vivo and ex vivo: ApoE^−/−^ on HD; In vivo: effect of statins on the atherosclerotic lesions in ApoE^−/−^ on HD	[101,102]
Ultrasound	Microbubbles	Monoclonal antibody	In vivo: ApoE^−/−^ on HD	[103]
Ultrasound	Microbubbles with magnetic streptavidin and fluorescent dye (1,1′-Dioctadecyl-3,3,3′,3′-tetramethylindocarbocyanine perchlorate, Dil)	Monoclonal antibody	In vivo: ApoE^−/−^ on HD	[104]
Ultrasound	Microbubbles	Antibody against VCAM-1 and ICAM-1 and synthetic polymeric sialyl Lewis X	In vivo: ApoE^−/−^ on HD	[105]
Ultrasound	Microbubbles	Nanobody (cAbVCAM-1-5)	In vivo: double knockout mice (LDL receptor and apoB48); Ex vivo: human carotid and femoral endarterectomy specimens	[106]
Ultrasound	ELIPs	Monoclonal antibody	In vivo: miniature pig on HD	[107]
Nuclear imaging	Fluorine-18	Tetrameric peptide	In vivo: ApoE^−/−^ on HD	[108]
Nuclear imaging	Iodine-123 and technetium-99 m	Peptide of the major histocompatibility complex-1 (MHC-1) molecule B2702	Ex vivo: Watanabe heritable hyperlipidemic rabbits; In vivo: murine model of plaque development induced by carotid artery ligation in ApoE^−/−^	[109,110]
Nuclear imaging	Technetium-99 m	Nanobody (cAbVCAM-1-5)	In vivo: ApoE^−/−^ on WD; In vivo: effect of statins on the atherosclerotic lesions in ApoE^−/−^ on WD; In vivo: effect of ezetimibe on the atherosclerotic lesions in ApoE^-/-^ on Paigen Diet	[111,112,113]
Nuclear imaging	Fluorine-18	Nanobody (cAbVCAM-1-5)	In vivo and ex vivo: ApoE^−/−^ on WD	[114]
Nuclear imaging	Fluorine-18	Nanobody (cAbVCAM-1-5)	In vivo and ex vivo: ApoE^−/−^ on WD	[115]
Nuclear imaging	Copper-64	Nanobody (cAbVCAM1-5)	In vivo: ApoE^−/−^ on HD and atherosclerotic rabbits	[116]
Raman spectroscopy	Biofunctional gold nanoprobe (BFNP) + Raman reporters	Antibody	In vivo: human adipose engraftment mouse model; Ex vivo: human coronary arteries	[117]

WD: Western Diet; HD: Hypercholesterolemic Diet.

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
