# Peer review of "VCAM-1 Target in Non-Invasive Imaging for the Detection of Atherosclerotic Plaques"

_biology, 2020, doi:10.3390/biology9110368_

Round 1

Reviewer 1 Report

I think "VCAM-1 Target in Non-Invasive Imaging for the Detection of
Atherosclerotic Plaques" is a very good paper.

Nevertheless, I'd like to suggest SOME MINOR REVISIONS to the authors.

  • English language is quite good but the authors may control the form of some words (i.e. in the abstract, characterised vs characterized) in order to satisfy English journal's rules
  • References are not always in accordance with authors' rules for the journal. PLEASE CHECK
  • Moreover, references are good and sufficiently comprehensive, but the authors should consider reading of following articles for eventual improvement and/or brief enlargement of the current DISCUSSION (average/good)
    • Molecules. 2020 Apr 16;25(8):1838. doi: 10.3390/molecules25081838.
    • J Nucl Med. 2020 Sep;61(9):1355-1360. doi: 10.2967/jnumed.119.238782. Epub 2020 Feb 28.

Best regards,

LG

Reviewer 2 Report

The authors describe in their review the various molecular tracers that are available for imaging of the biomarker VCAM-1 in the context of atherosclerosis. 

The review is build up as follows:

1) description of the disease development in atherosclerosis

2) extensive section on the role and structure of the biomarker VCAM-1

3) a short introduction on the imaging techniques currently used in clinic as well as on the different molecular imaging techniques that could be used

4) an extensive overview of the molecular tracers targeting specifically VCAM-1. 

In se the review is comprehensive, but the text is merely an enumeration of the different tracers that have been published, without critical assessment. I suggest the authors to review the manuscript thoroughly and address/discuss following points:

  • what are the advantages and disadvantages of the different molecular imaging techniques? Can they interchangeably be used in all patients, or are certain techniques better suited to specific patient groups?
  • Discussion on the exact indications for vcam-1 imaging. Is it for diagnosis, treatment follow-up or risk stratification? How sure are we that vcam-1 imaging will be of value in patients (or do we still need to demonstrate this?) How is this supported by literature data? 
  • which tracers are most promising? Why have they not yet been translated to the clinic? Which tracers have limitations for clinical use?
  • what is the clinical relevance of multimodel tracers?
  • ...

Some more general comments:

  • While the text contains many information on all the molecules involved in the pathophysiology of atherosclerosis, figure 1 and 2 are scarce in details
  • The review concerns imaging of VCAM-1, but not a single example is included. I suggest to make a compilation of the most promising data
  • the authors should avoid lone sentences or paragraphs consisting of only 1 sentence (e.g. pg 4). The manuscript should be revised for inconsistencies in abbreviations (VLA-4 is already used before, but only defined on pg 4, NIRF idem) and correct nomenclature for radiopharmaceutical should be used (e.g. mass number always in superscript, use of brackets, etc (https://static1.squarespace.com/static/59bd4d82d7bdce156a52b6bd/t/59bd51fd3bb053f5fd2f2c89/1494418606877/9May2017Consensus-nomenclature-rules.pdf)  
  • The authors should make a more clear distinguishment between the label that provides contrast for a specific imaging modality and the targeting moiety that specifically recognise VCAM-1. (in particular pg 7, line 272-274)
  • Table 1 needs to be improved 
    • since it concerns VCAM-1-targeting tracers, there is no need to repeat that for each probe in the table. Columns 'probe' and 'VCAM-1-specific probe's are thus often redundant. I would suggest to replace both by the contrast agent and the targeting moiety used. 
    • Name the same compounds in a consistent manner (e.g. cAbVCAM1-5 nanobody is used as basis for many different tracers; MB >< micro bubble)
    • Convert the table to landscape view to make it more readable --> more place for column 'types of study'

More specific comments:

  • Pg 7 line 233: FLuorophores for non-invasive, NIRF imaging, confocal microscopy and flow cytometry --> these are preclinical tools that are nice for research purposes, but are not relevant for clinical use. 
  • Pg 7 line 262: the statement that non-invasive NIRF is fast and inexpensive only holds true for mice, but irrelevant for patients as it cannot be applied. This should be made clear. 
  • Pg 7 line 265: Intravascular NIRF imaging uses protease-activated fluorescence agents. --> any fluorescent agent can be used, not only protease-activated ones
  • Pg 7 line 266: Not sure what the point is of that last sentence. It has no connection with the text above. 
  • Pg 7 line 271: For PET and SPECT imaging, there is no need for having an accessible intravascular target. This is only necessary for targeting vehicles that have difficulties with extravasating (such as particles for MRI or CT)
  • Pg 9 line 282: what is the clinical relevance of having a tracer that images both VCAM-1 and P-selectin simultaneously? Although more sensitive, will it still be specific enough? What Is the information that will be obtained? (see comment above, for which indication is VCAM-1 imaging exactly needed?)
  • Pg 9 line 284-286: Coupling [18F]NaF PET imaging and MPIO-VCAM-1 MRI opens new perspectives --> please elaborate. Which perspectives exactly? 
  • Pg 9 line 289: USPIO circulate longer than MPIO. Is this thus an advantage or not? Then follows a whole part on the potential of untargeted USPIO particles, but this should be moved to another section because it has nothing to do with VCAM-1 imaging. 
  • Pg 9 line 294: What is the difference between the study of Koch and Schmitz? Please provide more details on these studies 
  • Pg 9 line 302: Not clear to me whether the VCAM-1 targeted USPIO's perform better than the untargeted USPIOs?
  • Pg 9 line 309: what is the magnetic streptavidin used for? Please explain
  • Pg 10 line 317: please compare micro bubbles and ELIPs. Why should I chose for one or the other? 
  • Pg 10: please make clear that all the studies with nano bodies have been performed with the same cAbVCAM1-5 nano body, but labeled with a different contrast agent. 
  • Pg 10 line 344: Nanobodies are not nanoparticles!!!

Reviewer 3 Report

Thayse et al. present here an comprehensive review about imaging techniques for Atherosclerosis. The authors seem to be very familiar with the subject matter and have worked out essential points how imaging can and should be used in vascular diseases and where there are still possibilities. While the introduction and paragraphs about Atherosclerosis and Imaging Techniques are very well written, some weaknesses of the manuscript limit my enthusiasm.

Major points:

Some of the additions often seem pointlessly attached to good paragraphs. For example, line 266: is without any binding to the text. The authors should, however, read the text as a whole again and please avoid such jumps/breaks in the text flow.

The paragraph about VCAM-1 is mostly unnecessary and unfortunately not well written. It is actually just an unreadable list of quotations. I would not regret omitting lines 99-168. Stylistically this enumeration reminds me more of wikipedia. In addition, this part is not mentioned in the abstract. This shows me that even the authors see no necessity for a biochemical and molecular description of VCam-1 as protein, gene and the like.

Line 262-262: The sentences seem to be in wrong order. Since, NIRF is presented in the second sentence.

Reviewer 4 Report

In the current review, the authors have done a great job in reviewing how VCAM-1 can be utilized as a target in detecting atherosclerosis by invasive and non-invasive imaging methods.

Firstly, the authors thoroughly introduced the development of atherosclerosis in six stages illustrated by very nice figures. Then, introduced the role of VCAM-1 and its signaling pathways involved in the development of atherosclerosis. Secondly, the authors summarized invasive and non-invasive imaging modalities for atherosclerosis detection, mostly are already utilized in clinical practice. Then, the authors nicely summarized the VCAM-1 targeted imaging agents with a very informative table, and most of them still remain in laboratory models.

Overall this is a very informative review discussing the development of targeting VCAM-1 for atherosclerosis detection by various imaging modalities.

One major suggestion is, while authors only mentioned the clinical applicability of targeting VCAM-1, but there isn’t an apparent discussion on the possibility and applicability of targeting VCAM-1 on atherosclerosis detection. Can the authors provide more information on that? Some minor concerns are, 1. please organize some one-sentence paragraphs into more compact paragraphs following the logic flow. 2. Line 43, please give the full name of the “SHAPE” working group.

Round 2

Reviewer 2 Report

I just noticed that the request to review the manuscript was cancelled, though I just completed the review of the revised manuscript.

I am very sorry I could not do it earlier, but I would like to remember you that we do sometimes have other obligations that have priority. I accepted to review the revision because I think it is a logical thing to do if I reviewed the first version, but the timelines that you proposed were extremely short, especially since it was a major revision of the manuscript. If you want us to submit a proper review that will lead to a better article, this should be taken into account. It is not only about statistics on how fast a response is given.

Anyway, I think the manuscript was significantly improved and can be accepted. The authors should just pay attention to typo's and mention the long form of a word only at the first use with the abbreviation in between brackets.

E.g.

Table pg 10: should vbe iodine 123 instead of iodine 23

Pg 12 line 328: a dual-MPIO (DT-MPIO) --> should be dual -MPIO

Pg 13: line 378: IRM should be MRI